# Manufacturing Pitch and Polyethylene Blends-Based Fibres as Potential Carbon Fibre Precursors

**DOI:** 10.3390/polym13091445

**Published:** 2021-04-29

**Authors:** Salem Mohammed Aldosari, Muhammad A. Khan, Sameer Rahatekar

**Affiliations:** 1Enhanced Composite and Structures Centre School of Aerospace, Transport, and Manufacturing, Cranfield University, Cranfield MK43 0AL, UK; 2National Centre for Aviation Technology, King Abdulaziz City for Science and Technology (Kacst), Riyadh 11442, Saudi Arabia; 3Centre of Life-Cycle Engineering and Management School of Aerospace, Transport, and Manufacturing, Cranfield University, Cranfield MK43 0AL, UK; Muhammad.A.Khan@cranfield.ac.uk

**Keywords:** pitch, polyethylene, carbon fibres, extrusion, blend

## Abstract

The advantage of mesophase pitch-based carbon fibres is their high modulus, but pitch-based carbon fibres and precursors are very brittle. This paper reports the development of a unique manufacturing method using a blend of pitch and linear low-density polyethylene (LLDPE) from which it is possible to obtain precursors that are less brittle than neat pitch fibres. This study reports on the structure and properties of pitch and LLDPE blend precursors with LLDPE content ranging from 5 wt% to 20 wt%. Fibre microstructure was determined using scanning electron microscopy (SEM), which showed a two-phase region having distinct pitch fibre and LLDPE regions. Tensile testing of neat pitch fibres showed low strain to failure (brittle), but as the percentage of LLDPE was increased, the strain to failure and tensile strength both increased by a factor of more than 7. DSC characterisation of the melting/crystallization behaviour of LLDPE showed melting occurred around 120 °C to 124 °C, with crystallization between 99 °C and 103 °C. TGA measurements showed that for 5 wt%, 10 wt% LLDPE thermal stability was excellent to 800 °C. Blend pitch/LLDPE carbon fibres showed reduced brittleness combined with excellent thermal stability, and thus are a candidate as a potential precursor for pitch-based carbon fibre manufacturing.

## 1. Introduction

Carbon fibres are widely used in many industries, including aerospace, defence, construction, and healthcare, because of their high mechanical, thermal and electrical properties. However, fast expansion of the application of carbon fibres to industry will only continue if they can be produced at low cost without compromising physical properties [1]. Carbon fibres using synthetic rayon as the precursor have been developed since 1960 [2,3], and carbon fibres from pitch have been produced with a high elastic modulus since 1963 [4]. Carbon fibres derived from pitch precursors are primarily categorised into two kinds on the basis of their properties and type of pitch precursor. They are either mesophase pitch-based carbon fibres with high modulus and tensile strength and, in this study, with a molecular weight around 2600 g/mol [5], or isotropic pitch-derived carbon fibres with desirable mechanical properties. Isotropic pitch manufacture is not easy owing to the sudden appearance of mesophase spheres above a certain temperature [6]. The ease with which the mesophase appears is one of the reasons that makes mesophase pitch-based carbon fibre (MPCF) a very promising candidate and a target material for extensive research [7,8]. Other reasons include the occurrence of a high degree of anisotropy with greater alignment within the fibres and, hence, higher values of mechanical properties than those of polyacrylonitrile (PAN) or isotropic pitch-derived carbon fibres [3]. However, despite being a cheap precursor, purification of the pitch is quite expensive, and its final performance depends strongly on the extent of the defects introduced during processing. Manufacturing carbon fibres using pitch, which, along with synthetic polymers, is abundant, should reduce material costs, allowing for the wider use of carbon fibres. When compared to the high cost of PAN-based fibres, the mechanical performance of such cheap and readily available materials, such as lignin, has been explored but has been found to be inadequate [9], while carbon fibres are lightweight with excellent mechanical properties [10]. Textile-grade polyethylene is chemically simple and attractive as a precursor for carbon fibre production. It has the potential to reduce manufacturing costs by 50% relative to the PAN [11,12,13]. The PE is less expensive than PAN, because 50–65% of production costs are attributed to the synthesis of the PAN precursor, and that is not required in the case of PE. PE production via fusion spinning is cheaper, faster, more energy-saving and more environmentally friendly. PE has a higher carbon content and carbonation rate in addition to its congruence nature [11,13,14]. Its inherent defects make MPCF brittle, and that makes the spinning of these fibres difficult [3,4]. The stabilisation of MPCF is the most important stage in production, but it is so slow that it is by far the most inefficient stage [15].

Huang refers to a patent that describes a fresh technique of how melt-spun, sulphonated and carbonised polyethylene (PE) precursor could be used to manufacture carbon fibres with a yield of 75% [16]. Subsequently, there has been much research to develop low-cost PE-based carbon fibres with good mechanical properties, good compatibility, and high carbon content and carbonisation ratio precursor [11,13,17,18,19,20,21,22,23,24].

LLDPE has a high molecular weight, in the range of 50,000 to 200,000, which is higher than the molecular weight of both high-density and low-density polyethylene (HDPE and LDPE), and LLDPE also has a higher carbon content [25]. Our final aim is to develop carbon fibre-based PE and mesosphase pitch, where the pitch’s lower molecular weight of 400 to 800 may help to improve the amount of carbon in the carbon fibre [19,20,26,27].

To overcome the limitations imposed by poor spinnability due to the brittle nature of mesophase pitch, we use mixtures of mesophase pitch and LLDPE to reduce the brittleness of the pitch precursor fibres and to improve the fibre’s spinnability. LLDPE offers high ductility and can be formed into carbon fibres, hence it could be an excellent blending material with which to prepare mesophase pitch carbon fibre precursors with reduced brittleness. The characteristic behaviour of both MPCF and PE carbon can be improved by innovatively producing MP/PE-derived carbon fibres with the benefits of low cost, improved ductility, and good mechanical properties with minimum imperfections. In a review paper by Salem et al., it was shown that the aforementioned objectives could be realised using a Pitch/PE precursor blend to manufacture carbon fibre precursors with lower brittleness [1]. Research on PE/pitch carbon fibres is still in an early stage. This paper aims to contribute to that research by reporting an investigation of the properties of carbon fibre precursors manufactured from blends of mesophase pitch and PE with the goal of reduced brittleness. We can show that the brittle nature of the mesophase pitch can be reduced by adding a small fraction of LLDPE. Such fibres with lower brittleness will offer potentially better precursors for future carbon fibre manufacturing.

## 2. Methodology

### 2.1. Material

Precursors used in this study were LLDPE with density 0.918 g/cm^3^, melting point 121 °C, and softening point 99 °C, which was purchased from Sabic, mesophase pitch having density 1.425 g/cm^3^, melting point 298 °C, and softening point 268 °C, and mesophase pitch in which the content was mesophase-92%, which was received from Bonding Chemical.

### 2.2. Materials Processing

Fibre melt spinning was carried out using a single screw extruder (Noztek Pro Filament Extruder), with a bespoke screw design, swappable barrel, and filament tolerances of 1.75 ± 0.04 mm. The nozzle diameter was 0.5 mm, with a length of 1.5 mm. Pitch/PE pellets were fed into the machine in different ratios to obtain varying proportions of pitch blends. The extruder was operated at 315 °C, at an extrusion speed of 2.5 m/min, and a rotating winder was used for cold stretching as shown in Figure 1. When collecting the fibres, the stretching speed was fixed at 2.5 m/min [28]. The processing steps are shown in Figure 2. The designation of samples with respect to varying proportions of pitch and LLDPE is shown in Table 1.

## 3. Characterisation Methods

### 3.1. Microscopy

#### 3.1.1. Optical

A Nikon ECLIPSE ME600 optical microscope was used to determine the fibre diameter; the optical images were recorded at 20× magnification. The microscope can be used in differential interference contrast (DIC), which is a method of imaging based on contrast difference of samples.

#### 3.1.2. Scanning Electron Microscopy (SEM)

The microstructure of prepared samples was analysed using a scanning electron microscope, a Tescan VEGA 3 with Oxford Instruments EDS software (Aztec). Prior to analysis, the Quorum 150T ES sputter coater was used to coat a gold layer onto the sample. The samples were prepared by cutting lengths with a clean blade and mounting both vertically (to image end-on) and horizontally (to image surface features) onto aluminium SEM stubs with double-sided carbon tabs. The gold-coated samples were then loaded into the SEM chamber separately due to the height difference. Images were taken at the magnifications noted on the data bar; typically, 500×, 1000×, and 4000×. EDS analysis was conducted to distinguish the pitch from the polyethylene. Measurements were taken of the diameter on the horizontally mounted fibres.

### 3.2. Static Mechanical Analysis (Tensile Test)

Tensile testing of melt-spun MP/PE fibres was performed using a DEBEN Microtest fibre tensile tester coupled with Leica EC4 Microscope; the test was conducted according to Standard ISO 11,566, 1996 Figure A1: Carbon fibre—Determination of the tensile properties of single-filament specimens. Six tests were carried out for each sample, and the repeatability of the tests was confirmed. Stress and strain were calculated using Equations (1) and (2).
(1)$$Stress\;\left(\sigma \right)=\frac{F}{A_{c}}=\frac{F}{\pi \frac{d^{2}}{4}}$$
(2)$$Strain\;\left(\epsilon \right)=\frac{Change\;in\;length\;}{Orignal\;length}=\frac{Elongation}{Guage\;Length}=\frac{\Delta L}{L_{0}}=\frac{L-L_{0}}{L_{0}}$$
*d* = diameter of fibres (mm), *F* = Axial load (N).

### 3.3. Differential Scanning Calorimetery (DSC)

A Mettler Toledo Differential Scanning Calorimeter, DSC Q 2000, was used to perform the DSC analysis. This equipment was used to measure the properties of prepared samples. The samples were heated from −50 °C to 200 °C at a rate of 20 °C/min under a nitrogen atmosphere, and the samples were held at 200 °C for 5 min to eradicate their previous thermal history. Nonisothermal behaviour and kinetics were investigated by cooling these samples at a rate of 20 °C/min. The results were obtained after the heating and cooling cycles. The crystallinity of the samples was measured using the following equation:(3)$$\chi \left(\%\right)=\frac{Enthalpy\;of\;fusion\;\left(\Delta H\right)}{Enthalphy\;of\;fusion\;for\;Matrix\left(\Delta H^{0}\right)}\times 100$$
χ=Crystallinity of the system.ΔH=Enthalpy of fusion of LLDPE and Pitch fibres.$\Delta H^{0}=Enthalpy\;of\;fusion\;of\;neat\;Pitch.$

### 3.4. Thermogravimetric Analysis (TGA)

A Mettler Toledo, Thermogravimetric Analysis TGA Q 500, was used to perform the TGA analysis; to measure the properties of the prepared samples. The samples were heated from 50 to 800 °C at a heating rate of 20 °C/min under a nitrogen atmosphere, and the samples were held at 800 °C for 5 min to eradicate their previous thermal history.

## 4. Results and Discussion

### 4.1. Microscopy

#### 4.1.1. Optical Microscopy of Fibre

It was observed that the different samples produced different fibre diameters, which depended upon the LLDPE content. The fibre diameter decreased with an increase in LLDPE content. This is shown in Figure 3 where fibre diameter is seen with LLDPE content in LLDPE/mesophase pitch blends as shown in Table 2.

We see from Figure 3 and Figure 4 and Table 2 that the fibre diameter decreases with increasing LLPDE content. The diameter of the fibre varies due to the change in the applied axial elongation force required during the drawing process due to the change in LLDPE content. It has previously been reported [28] that the morphology of the blends is changed by varying the content of the blend’s partner.

The diameter of fibres in the extrusion process is greatly affected by many factors, such as extrusion speed, the viscosity of the polymers, the diameter of orifice, stretching or drawing ratio, and speed of extrusion. The optical microscope was used to determine the diameter of the cold-drawn samples, and optical microscope images of these with respect to LLDPE content are presented in Figure 3, which shows that different samples produced different fibre diameters, which decreased with an increase in LLDPE content.

#### 4.1.2. Scanning Electron Microscopy of Fibres

Figure 4 shows the morphology of a range of LLDPE/mesophase pitch blends. The SEM image shows that pitch fibres have a tendency to form microfibrils inside the fibres; Figure 4a [29]. As the LLDPE content is increased, there is a tendency for microfibre formation inside the blend fibres to reduce; see Figure 4b–e. Post adding of LLDPE with pitch, a decrease in the cluster formation and low alignment is observed in the mesophase pitch. The pure/neat LLDPE fibre does not show any microfibrous pitch (silver), only LLDPE (black solid). Therefore, we can conclude that the relative content of LLDPE has a very significant effect on fibre morphology and diameter. In mesophase pitch fibre fabrication, the molten liquid crystalline mesophase can be oriented within the die by shear treatment during a soften-spinning process. Overall, the shear forces involved lead to a strongly oriented structure, with the mesophase domains elongated in cylinders parallel to the axis of the fibres (circumferential fibres). The microstructural study suggests that the radial orientation of the fibres obtained from the mesophase pitch originates from the flow to throw across the die [30]. The spinning stage is the most important stage in the manufacture of mesophase pitch blend LLDPE, to control the shape, transverse texture and orientation of the fibre, which results in higher performance.

The mesophase pitch shows the liquid crystalline phase and preferential orientation of the pitch molecules, giving appropriate reference of the liquid crystal and flow orientation behaviour of pitch. The flow orientation behaviour may be causing microfibre formation. The polyethylene does not show the liquid crystalline behaviour, hence, when you add polyethylene in pitch, it decreases the microfibre content in pitch, and when you extrude the neat polyethylene, there are no microfibres due to lack of liquid crystal behaviour.

Pitch microfibers formation occurs because of the liquid crystalline mesophase nature of the pitch. It allows alignment and therefore microfiber formation, LLDPE not liquid crystalline, therefore does not make microfiber.

### 4.2. Tensile Tests of LLDPE/Mesophase Pitch Blends

LLDPE is inexpensive, flexible, and is extensively employed in several different forms. One study has shown that the tensile strength of LLDPE is typically in the region of 6.1 MPa and that the tensile strength can be increased by the addition of straw fibres [31]. Another study has shown the tensile strength of LLDPE can reach 9.9 MPa [32]. Others have reported that the tensile strength of LLDPE can be extended to 22 MPa, by the inclusion of multiwalled carbon nanotubes [33]. These improvements in the properties of LLDPE suggest extending its possible use in combination with new applications.

To understand the effect of extrusion on static mechanical properties, two types of extruded strands were used for tensile testing. These strands were related to the number of the extrusion cycle, in which sample-one is the designation for a single extruded strand, and sample-two is the designation for double extruded strands. Tensile tests were performed for both sample-one and sample-two strands. It was found that the tensile strength and tensile modulus were different for sample-one and sample-two. For sample one, the extruded tensile strength and modulus were 3.30 MPa and 151 MPa, respectively, and for the double extruded strands, the respective values were 3.06 MPa and 21.42 MPa. It was clearly seen that both tensile modulus and fracture elongation were higher for sample-one in comparison with sample-two. However, tensile strength was very slightly higher for sample-two in comparison with sample one, and from this we conclude that sample-two has a lower modulus and elongation at failure in comparison with sample-one due to degradation and damage that occurred in the LLDPE chains during the second extrusion. This damage may be due to the higher temperature and shearing during mixing in the extrusion process [34,35,36].

Figure 5a shows the stress vs. strain curve of pure extruded LLDPE. It can be seen that the curve is smooth, and tensile strength is typically in the range of 40 MPa, which indicates that the LLDPE plays a key role in load-bearing as well as toughening materials. The effect of increasing the proportion of LLDPE on the brittle mesophase pitch can be seen in Figure 5b, where the mechanical strength of LLDPE/mesophase pitch blends comes from both LLDPE and mesophase pitch.

Figure 5b shows the stress vs. strain curve of LLDPE/mesophase pitch blends for neat/pure pitch (0 wt%), 5 wt%, 10 wt%, 15 wt% and 20 wt% of LLDPE. It can be seen that the samples show different stress vs. strain behaviour depending on the LLDPE content. The lowest tensile strength (1.0 MPa) and lowest strain (5%) were found in the case of neat mesophase pitch. The tensile strength and strain increased steadily with LLDPE content up to the maximum of 20 wt% (10.0 MPa) and maximum strain (25%). We define brittleness as low strain to failure, therefore the pitch fibre sample shows brittle behaviour. When the LLDPE content increased, the strain to failure of these samples increased; thus, adding LLDPE to the fibres reduces brittleness. Table 3 shows a summary of the tensile strength, tensile modulus, and strain to failure of the neat pitch fibres and blends of pitch and LLDPE-based fibres along with the standard deviation for each type of fibre. It can be seen that, for the prepared samples, the higher the LLDPE content, the greater the tensile strength, tensile modulus, and strain at failure.

### 4.3. Differential Scanning Calorimetery for LLDPE/Mesophase Pitch Blends

The crystallization of several LDPE and LLDPE blends has been evaluated using differential scanning calorimetry (DSC). It was observed that in the region between 110 and 120 °C, where pure LDPE does not melt, there was an increase in the population of crystallites melting. A reduction in the crystallite population across the range where LDPE shows its primary melting peaks (90–110 °C) has been observed, suggesting that a proportion of the lamella had shifted towards a higher melt temperature [37]. DSC has shown that blends of LDPE and LLDPE show different single melting peak endotherms for different compositions in the range 10 to 90 wt% LLDPE [38].

Kinetic data were acquired by least-squares analyses of experimental points obtained by differentiating primary thermograms. Degradation was found to be higher for LDPE than for LLDPE. Degradation has been seen as a primary order for LDPE and second order for LLDPE [38].

The thermal behaviour of LLDPE/mesophase pitch, over the range 0 wt% LLDPE/mesophase pitch to 20 wt% LLDPE/mesophase pitch, was investigated using DSC. The DSC experiments were carried out over a range of melting and crystallization temperatures of LLDPE to avoid any degradation of the LLDPE.

Figure 6a shows the crystallization behaviour of prepared samples, and it is noticed that neat mesophase pitch does not show any marked variation in crystallization behaviour, with no peaks or troughs occurring on the plot [39].

In Figure 6a, we see that neat LLDPE shows a well-organised crystallization peak and temperature. We also see that adding different proportions of LLDPE to the LLDPE/mesophase pitch blend clearly affects the crystallization behaviour of the blend, both the crystallization temperature and the enthalpy of crystallization (∆H_c_, area under the exotherm curve). The enthalpy of crystallization for LLDPE/mesophase pitch blend increases with the proportion of LLDPE. Figure 6 shows that the crystallization temperature of LLDPE/mesophase pitch blend decreases with an increase in the proportion of LLDPE, which also indicates that the crystallization temperature of blends is dependent upon the LLDPE content.

Similarly, the melting temperature and enthalpy of fusion of the samples change with the proportion of LLDPE in the LLDPE/mesophase pitch blend, see Figure 6b, and Table 4. The melting temperature increases with the proportion of LLDPE in the blend, so that the maximum melting temperature is for pure LLDPE. The enthalpy of fusion (∆H_m_, the area under the endotherm) is highest for pure LLDPE; this is due to the absence of crystalline domains in pure LLDPE compared to LLDPE/mesophase pitch blends. We see that the presence of LLDPE in LLDPE/mesophase pitch blends affects the melting and crystallization behaviour of the blends.

### 4.4. Thermogravimetric Analysis of LLDPE/Mesophase Pitch Blends

The objectives of the thermogravimetric analysis (TGA) tests were to measure the thermal decomposition of several blends of LLDPE/mesophase pitch as they burned in an air environment. Analysis of the thermal stability of a material is generally done by TGA and derivative TGA. TGA was also used for the thermal degradation of the composites. A higher decomposition temperature indicates greater thermal stability of the material [33,40].

TGA is used to measure the rate and amount of weight transformation in the material, either as a function of temperature or isothermally as a function of time, in a controlled atmosphere. It can be used to characterise any material that exhibits a weight change during combustion and to discover phase changes due to decomposition, oxidation or dryness. This data helps classify the percentage weight transformation and chemical structure, handling, plus end-use performance.

The TGA was carried out in the temperature range of 100 to 800 °C. The thermal degradation of neat LLDPE and different wt% LLDPE/mesophase pitch fibres was studied, and typical thermal stability curves are shown in Figure 7 for a range of mixes from neat mesophase pitch to neat LLDPE.

Figure 7 shows the weight loss vs. temperature of prepared samples, and it is seen that TGA shows the complete decomposition of the LLDPE at 530 °C. However, the 20 wt% LLDPE/mesophase pitch showed only 20% decomposition at 530 °C, confirming that mesophase pitch has greater stability than neat LLDPE, though the proportion of LLDPE will affect the thermal stability of an LLDPE/mesophase blend. By blending different proportions of LLDPE with mesophase pitch, the decomposition temperature of different LLDPE/mesophase pitch blends was found; see Figure 7. The fibres containing over 5 wt% and 10 wt% LLDPE could retain over 70 wt% of the fibre mass even at 800 °C. These two samples with reduced brittleness and high thermal stability can potentially offer excellent precursors for manufacturing carbon fibres.

## 5. Conclusions

The aim of this study was to reduce the brittleness of pitch precursor fibres. These fibres are commonly used for manufacturing carbon fibres blended with a small fraction of LLDPE. We have demonstrated a simple but effective method of blending mesophase pitch with 5% to 20% (i.e., weight percentage) of LLDPE. The addition of LLDPE showed an increase of strain to failure value up to 25%, hence increases ductility. Thus, these blends reduce brittleness in all the LLDPE/pitch fibres. The LLDPE/pitch blend fibres also showed improved tensile strength and Young modulus from 1.3 to 10.3 MPa and from 428 to 763 MPa, respectively. The precursors used in this study along with blend can be used to produce potential carbon fibres with superior mechanical properties.

Morphology observed using SEM analysis revealed that neat LLDPE fibres did not show any microfibrils, whereas the neat pitch fibres showed microfibrillated structure. As the LLDPE content was increased, the microfibrous formation was seen to decrease, until with 100% LLDPE no microfibrous areas were observed and the appearance was close to solid black.

DSC analysis showed crystallisation temperature of LLDPE in the pitch/LLDPE blend ranged from 99 to 103 °C, and the melting temperature was in the range 120 to 124 °C.

TGA analysis showed high thermal stability, making the obtained pitch/LLDPE fibre precursor suitable to develop less brittle carbon fibres for high-performance composites, particularly for use in the aerospace sector.

A comprehensive investigation of the PE/pitch morphology is still required. It is key to understand how morphology can be controlled during fabrication as it affects the crack resistance and ductility of the obtained fibre. The stabilisation, carbonisation and graphitisation of PE/pitch precursors need to be explored to increase the possibility of producing more ductile carbon fibres. The carbonisation step, which deals with the presence of polyethylene chain, needs to be explored further. It can potentially increase the bent and loop regions of the graphitic planes and hence reduce the crack formation in the fibre.

## Figures and Tables

**Figure 1 polymers-13-01445-f001:**
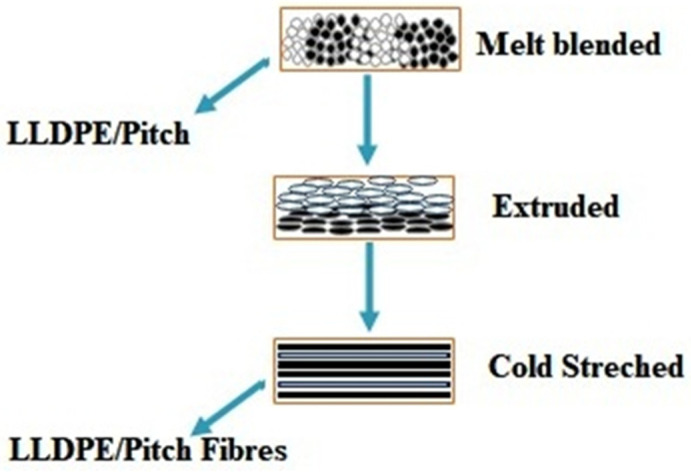
Production of fibres from melt blends.

**Figure 2 polymers-13-01445-f002:**
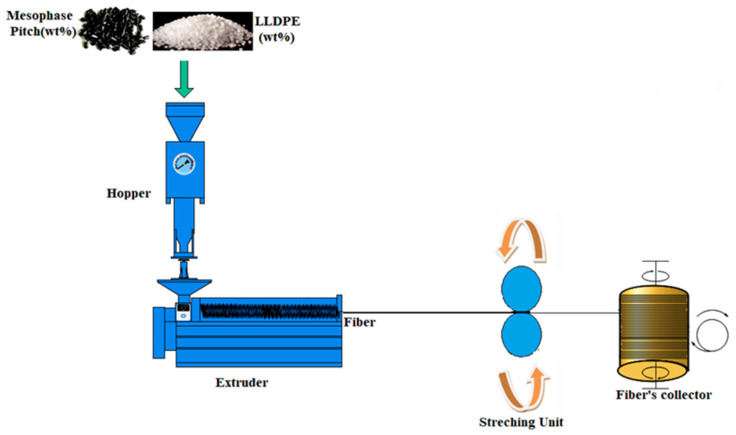
Schematic diagram of material processing.

**Figure 3 polymers-13-01445-f003:**
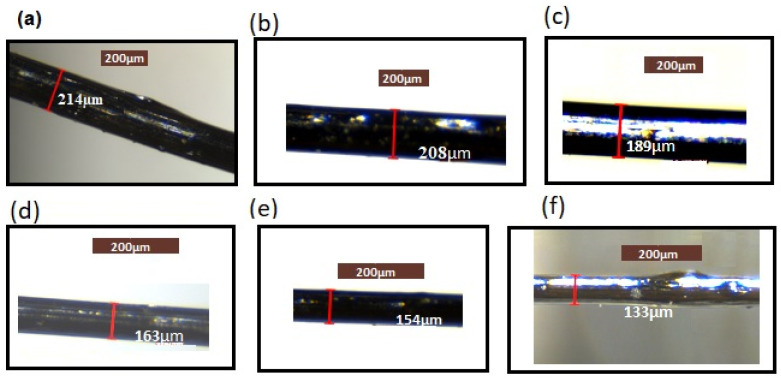
Optical microscope image of fibres with different LLDPE content. (**a**) Mesophase pitch, (**b**) 5 wt% LLDPE/mesophase pitch (**c**) 10 wt% LLDPE/mesophase pitch, (**d**) 15 wt% LLDPE/mesophase pitch, (**e**) 20 wt% LLDPE/mesophase pitch and (**f**) LLDPE.

**Figure 4 polymers-13-01445-f004:**
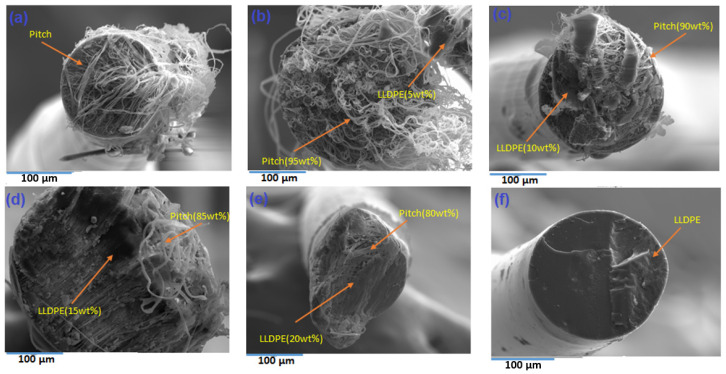
Morphology of fibres with LLDPE content in LLDPE/mesophase pitch blend. (**a**) Pure mesophase pitch, (**b**) 5 wt% LLDPE/mesophase, (**c**) 10 wt% LLDPE/mesophase pitch, (**d**) 15 wt% LLDPE/mesophase pitch, (**e**) 20 wt% LLDPE/mesophase pitch and (**f**) 100% LLDPE.

**Figure 5 polymers-13-01445-f005:**
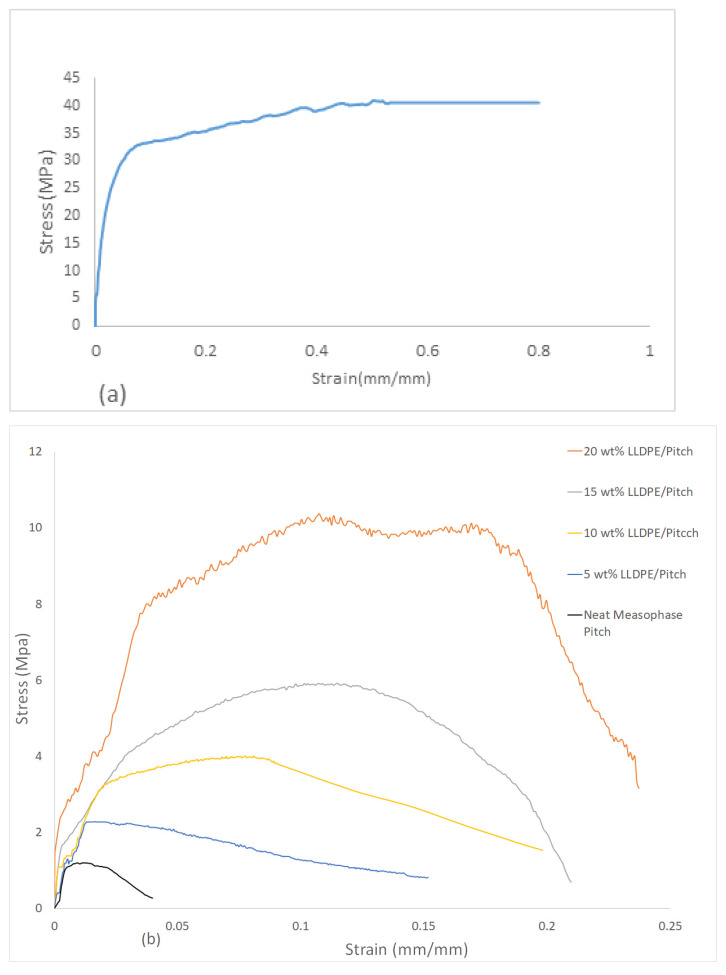
(**a**) Tensile strength of neat LLDPE; (**b**) Tensile strength of neat mesophase pitch fibres and LLDPE/mesophase pitch blend-based fibres with varying content of LLDPE.

**Figure 6 polymers-13-01445-f006:**
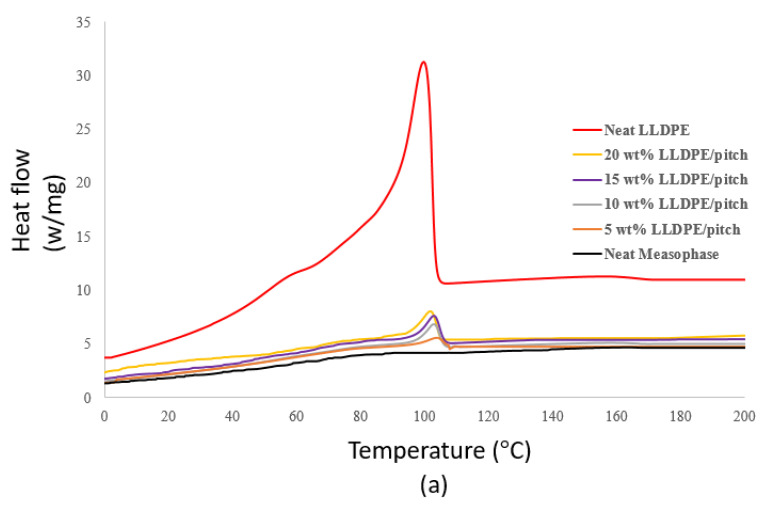
(**a**) Crystallisation, and (**b**) melting behaviour of LLDPE and LLDPE/mesophase pitch blend.

**Figure 7 polymers-13-01445-f007:**
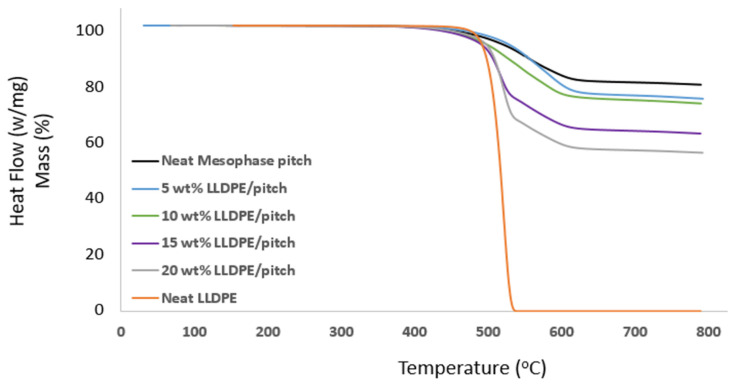
TGA curves of various proportions of LLDPE in LLDPE/mesophase pitch blends.

**Table 1 polymers-13-01445-t001:** Blends in terms of wt% mesophase pitch and LLDPE.

Blend Designation	Mesophase Pitch	LLDPE
Mesophase Pitch	100 wt%	0 wt%
LLDPE (5 wt%)/Mesophase Pitch	95 wt%	5 wt%
LLDPE (10 wt%)/Mesophase Pitch	90 wt%	10 wt%
LLDPE (15 wt%)/Mesophase Pitch	85 wt%	15 wt%
LLDPE (20 wt)/Mesophase Pitch	80 wt%	20 wt%
LLDPE (100 wt%)	0 wt%	100 wt%

**Table 2 polymers-13-01445-t002:** Blends in terms of wt% mesophase pitch and LLDPE fibres diameter.

Blend Designation	Fibre Diameter, µm
Mesophase Pitch	214 (±0.43)
LLDPE (5 wt%)/Mesophase Pitch	208 (±0.54)
LLDPE (10 wt%)/Mesophase Pitch	189 (±0.24)
LLDPE (15 wt%)/Mesophase Pitch	163 (±0.36)
LLDPE (20 wt%)/Mesophase Pitch	154 (±0.38)
LLDPE (100 wt%)	133 (±0.53)

**Table 3 polymers-13-01445-t003:** Tensile strength, modulus, and strain at failure of the prepared samples.

Samples	Tensile Strength (MPa)	Tensile Modulus (MPa)	Strain at Failure
Mesophase Pitch	1.38 (±0.26)	428 (±4.3)	0.03 (±0.021)
LLDPE (5 wt%)/Mesophase Pitch	2.23 (±0.34)	477 (±5.7)	0.15 (±0.023)
LLDPE (10 wt%)/Mesophase Pitch	4.01 (±0.43)	628 (±5.4)	0.19 (±0.022)
LLDPE (15 wt%)/Mesophase Pitch	5.90 (±0.58)	682 (±4.5)	0.21 (±0.024)
LLDPE (20 wt%)/Mesophase Pitch	10.3 (±0.87)	763 (±5.3)	0.23 (±0.025)
LLDPE (100 wt%)	40.0 (±0.98)	994 (±6.4)	0.80 (±0.022)

**Table 4 polymers-13-01445-t004:** Melting and crystallisation temperatures of LLDPE and LLDPE/mesophase pitch blend.

Samples	Melting Temperature (°C)	Crystallization Temperature (°C)	Enthalpy of Fusion (J/g) Sample	Enthalpy of Fusion (J/g) LLDPE
LLDPE 5 wt%/Mesophase Pitch	120	99	4	80
LLDPE 10 wt%/Mesophase Pitch	120.7	100	25	250
LLDPE 15 wt%/Mesophase Pitch	121	101	27	180
LLDPE 20 wt%/Mesophase Pitch	123	102	31	155
LLDPE 100 wt%	124	103	241	241

## Data Availability

Data can be available on a request from the corresponding author via email.

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
