# Peer review of "Manufacturing Pitch and Polyethylene Blends-Based Fibres as Potential Carbon Fibre Precursors"

_polymers, 2021, doi:10.3390/polym13091445_

Round 1
Reviewer 1 Report
This work studied the structure and properties of pitch/LLDPE blends for potential carbon fibre precursors. Although this work has achieved some valuable results, there are some problems that should be clarified and more experiments and discussion should be supplied before further consideration.
- It will be more valuable if the attempt for manufacturing the carbon fibres by using pitch/LLDPE blends can be achieved in the current work.
- In introduction, the authors should clarify the advantage of using LDPE to fabricate carbon fibre precursors compared to other polymers, such as PAN and cellulose.
- The loading of LDPE ranges from 5 wt% to 20 wt%. What about higher concentration? The authors should explain the optimization design of LDPE loading.
- The revolutions of Figure 3 and 5 are not high. Please improve the quality of these figures.
- There are some mistakes. For example, in conclusion (line 305), young module is a spelling error. It should be Young’s modulus. Please carefully check the whole manuscript.
Reviewer 2 Report
Carbon fibres derived from pitch precursors are primarily categorised into two kinds based on their properties and type of pitch precursor. To overcome the limitations imposed by poor spinnability due to the brittle nature of mesophase pitch we use mixtures of mesophase pitch and LLDPE to reduce the brittleness of the pitch precursor fibres and to improve the fibre’s spinnability. LLDPE offers high ductility and can be formed into carbon fibres, hence it could be an excellent blending material with which to prepare mesophase pitch carbon fibre precursors with reduced brittleness. The characteristic behavior of both MPCF and PE-carbon can be improved by innovatively producing MP/PE derived carbon fibres with the benefits of low cost, improved ductility, and good mechanical properties with minimum imperfections. In this manuscript, this paper reports the development of a unique manufacturing method using a blend of pitch and Linear Low-Density Polyethylene (LLDPE) from which it is possible to obtain precursors which are less brittle than neat pitch fibres. This study reports on the structure and properties of pitch and LLDPE blend precursors with LLDPE content ranging from 5wt% to 20wt%. Fibre microstructure was determined using Scanning Electron Microscopy (SEM) which showed a two-phase region having distinct pitch fibre and LLDPE regions. The topic is important, the results are interesting and the methodology followed is appropriate, while the content falls well within the scope of this Journal. In general the paper makes fair impression and my recommendation is that it merits publication in this Journal, after the following major revision:
- The introduction should be reconstructed to present a coherent literature review. It may help the authors by answering the following questions: Why are these works relevant? Which specific problems were addressed? How are the previous results related with the latest work? What are the outstanding, unresolved, research issues? Answering the questions leads to the novelty of the proposed work naturally.
- In Figs .6 and 7, the authors should give the explanations for the difference of data collected from different sources.
- Methodology part. Although the results look “making sense”, the current form reads like a simple lab report. The authors should dig deeper in the results by presenting some in-depth discussion.
- As the LLDPE content was increased, the microfibrous formation was seen to decrease until with 100% LLDPE and no microfibrous areas were observed and the appearance was close to solid black. The authors should give some explanation on above results and analyze the physical mechanism in detail.
- Carbon fibres have been widely used in the industry. Carbon fibres have been applied in a number of practical applications, for example fibrous porous media, (see [A fractal model for capillary flow through a single tortuous capillary with roughened surfaces in fibrous porous media, Fractals, 2021, 29(1):2150017; Fractals, 2019, 27(7): 1950116]). Authors should introduce some related knowledge to readers. I think this is essential to keep the interest of the reader.
- Please, expand the conclusions in relation to the specific goals and the future work.
Author Response
Dear Reviewer 2
We would like to thank you for your helpful comments. we attached response

Round 2
Reviewer 1 Report
The manuscript has been improved by the authors. It can be accepted for publication.
Reviewer 2 Report
In Ref. 8, “28(4): 1–11” should be corrected as “29(1):2150017”.